# Peptides Derived from the SARS-CoV-2 S2-Protein Heptad-Repeat-2 Inhibit Pseudoviral Fusion at Micromolar Concentrations: The Role of Palmitic Acid Conjugation

**DOI:** 10.3390/ijms25126382

**Published:** 2024-06-09

**Authors:** Nejat Düzgüneş, Zhihua Tao, Yuxia Zhang, Krzysztof Krajewski

**Affiliations:** 1Department of Biomedical Sciences, Arthur A. Dugoni School of Dentistry, University of the Pacific, 155 Fifth Street, San Francisco, CA 94103, USA; 2BPS Bioscience, 6405 Mira Mesa Blvd, Suite 100, San Diego, CA 92121, USA; ztao@bpsbioscience.com (Z.T.); yzhang@bpsbioscience.com (Y.Z.); 3Department of Biochemistry and Biophysics, 3057 Genetic Medicine, CB# 7260, University of North Carolina School of Medicine, Chapel Hill, NC 27599, USA; krzysztof_krajewski@med.unc.edu

**Keywords:** membrane fusion, membrane proteins, fusion inhibitor, six-helix bundle

## Abstract

SARS-CoV-2 S-protein-mediated fusion is thought to involve the interaction of the membrane-distal or N-terminal heptad repeat (NHR) (“HR1”) of the cleaved S2 segment of the protein and the membrane-proximal or C-terminal heptad repeat (CHR) (“HR2”) regions of the protein. We examined the fusion inhibitory activity of a PEGylated HR2-derived peptide and its palmitoylated derivative using a pseudovirus infection assay. The latter peptide caused a 76% reduction in fusion activity at 10 µM. Our results suggest that small variations in peptide derivatization and differences in the membrane composition of pseudovirus preparations may affect the inhibitory potency of HR2-derived peptides. We suggest that future studies on the inhibition of infectivity of SARS-CoV-2 in both in vitro and in vivo systems consider the need for higher concentrations of peptide inhibitors.

## 1. Introduction

The Severe Acute Respiratory Syndrome Coronavirus-2 (SARS-CoV-2) binds the angiotensin-converting enzyme 2 (ACE2) receptor on its host cells via the S1 domain of its spike protein, S (Figure 1) [1]. The transmembrane protease/serine subfamily (TMPRSS) cell surface proteases cleave the S-protein into S1 and S2 sub-fragments. The receptor-binding domain (RBD) of S is located in the N-terminal S1 domain of S. Similar to the membrane fusion activity of the Env protein of HIV-1, the fusion activity of the S-protein is associated with the C-terminal S2 domain.

S-mediated fusion is thought to involve the interaction of the membrane-distal or N-terminal heptad repeat (NHR) (also termed “HR1”) of S2 and the membrane-proximal or C-terminal heptad repeat (CHR) (termed “HR2”) regions of the protein.

Upon binding to its receptor, the S-protein undergoes a conformational change where the S1 subunit is released, the fusion peptide of the S2 subunit is inserted into the host cell membrane, and the HR1 and HR2 domains interact with one another, resulting in the formation of the “six-helix bundle.” This conformational change brings the viral and cell membranes into close proximity (Figure 2) [1].

The HR1 domain of SARS-CoV-2 has a high α-helicity and a high binding affinity to the HR2 domain. A peptide derived from the HR2 domain (“2019-nCoV-HR2P”) was found to inhibit cell–cell fusion induced by the SARS-CoV-2 S-protein, with an IC_50_ of 0.18 µM [3]. These authors also showed that this peptide interacts with a peptide derived from the HR1 region and forms a helical structure characteristic of the six-helix bundle, confirming the formation of the hairpin structure depicted in Figure 2. The same laboratory developed a similar peptide (“EK1”) derived from the HR2 region of the coronavirus OC43 and showed that it inhibits the fusion activity of multiple coronaviruses [4]. They subsequently reported that EK1 inhibits SARS-CoV-2 cell–cell fusion at an IC_50_ of 0.32 µM [5]. They also found that linking cholesterol to the C-terminus of EK1 through 5 additional amino acids and poly(ethylene glycol)_4_ lowered the IC_50_ from 286.7 nM to 48.1 nM. The palmitic acid derivative of EK1 (designated EK1P) lowered the IC_50_ for the inhibition of SARS-CoV-2 S-protein-mediated cell–cell fusion to 69.2 nM.

These observations indicate that synthetic peptides and their derivatives can be used as therapeutic agents against SARS-CoV-2. The sites of interaction of EK1 and possibly other peptides with the S-protein are shown in Figure 3.

We were intrigued by the very low inhibitory concentrations of EK1 and EK1P described by Xia et al. [3,5], and sought to test peptides of an identical amino acid sequence in a different, but analogous, experimental system. We synthesized “Peptide 1” as identical to EK1, except for the addition of the poly(ethylene glycol)_5_ linker (PEG5) at the C-terminus. Using a pseudotyped vesicular stomatitis virus expressing the SARS-CoV-2 S-protein, we tested the fusion inhibitory activity of Peptide 1. We also tested the palmitoyl derivative of Peptide 1, designated as “Peptide 2”, to compare with the antiviral activity of EK1P. The inclusion of the PEG5 in Peptide 1 was to be able to directly compare the palmitic acid derivative with the non-palmitoylated peptide. We envision the role of the palmitic acid as a means to possibly attach the peptide to the viral membrane or the cell membrane, facilitated by its hydrophobic character, thereby increasing the local concentration of the peptide near the target S-protein.

## 2. Materials and Method

### 2.1. Cells and Supplies

The SARS-CoV2-spike pseudovirus and the ACE2-HEK293 recombinant cell line were prepared in-house by BPS Bioscience (San Diego, CA, USA) (#79942 and #79951, respectively). The MEM medium (#SH30024.01), Na-pyruvate (#SH30239.01) and Pen-strep (#SV30010) were obtained from Hyclone (Cytiva, Marlborough, MA, USA). The fetal bovine serum was obtained from Life Technologies (#10082147) (Waltham, MA, USA). The non-essential amino acids (25-025-Cl) and 96-well tissue-culture-treated white clear-bottom assay plates (#3610) were from Corning (Corning, NY, USA). The puromycin was purchased from InvivoGen (#ant-pr-1) (San Diego, CA, USA). The Spike S1 Neutralizing Antibody (Clone#G10xA1) (#101326) and the ONE-Step Luciferase Assay System (#60690) were from BPS Bioscience.

### 2.2. Peptides

The peptides were synthesized at the UNC Peptide Synthesis Facility (RRID:SCR_017837) on a CEM Liberty Blue microwave peptide synthesizer (HE-SPPS methodology, [6]) using Fmoc-ProTide Rink LL amide resin (loading 0.2 mmol/g). The C-terminal lysine residue was introduced with an ivDde sidechain protecting group; this allowed for orthogonal deprotection of the ε-amino group of this residue (3 × 5 min, 5% hydrazine in N,N-dimethylformamide (DMF)) after the peptide sequence was fully synthesized and the N-terminal amine was protected with a Boc group. Subsequently, the ε-amino group of this Lys residue was coupled with Fmoc derivative of 17-amino-3,6,9,12,15-pentaoxaheptadecanoic acid (Fmoc-NH-Peg_5_-COOH), and after Fmoc deprotection, for Peptide 2, coupled with palmitic acid. Both coupling reactions were performed manually using hexafluorophosphate azabenzotriazole tetramethyluronium (HATU) as a coupling reagent in the presence of N,N-diisopropylethylamine in DMF and confirmed with ninhydrin tests. The peptidyl resin was washed 3 × with dichloromethane, 3 × with methanol, and dried in a vacuum chamber overnight.

The peptide was cleaved from the resin and deprotected by a 2 h incubation with 2 mL of cleavage solution (92.5% trifluoroacetic acid, 2.5% triisopropylsilane, 2.5% ethane-1,2-dithiol, 2.5% water) and precipitated by addition into cold diethyl ether (~30 mL). The precipitate was collected by centrifugation and triturated twice with 5 mL of diethyl ether. The residual ether was allowed to evaporate, and the peptide was dissolved in 50% acetonitrile and lyophilized. The crude peptides were purified by preparative reversed-phase high performance liquid chromatography (RP HPLC) on a Waters SymmetryShield RP18 column and lyophilized. The purified peptides were characterized by matrix-assisted laser desorption/ionization time of flight mass spectrometry (MALDI TOF MS) and analytical HPLC (Millipore (Burlington, VT, USA) Chromolith RP-18e column). 

### 2.3. Cell Culture and Assay Conditions

The ACE2-HEK293 cells were cultured in MEM medium with 10% FBS, 1% non-essential amino acid, 1 mM Na-pyruvate, 1% Pen-strep, and 0.5 μg/mL puromycin. The ACE2-HEK293 cells were seeded at 8000 cells per well into a white clear-bottom 96-well microplate in 90 μL of assay medium. Ten microliters of preincubated virus/compound mix was added into each well of the ACE2-HEK293 cells. For the control cells, the same number of the ACE2-HEK293 cells were seeded, but no virus or compound was added. The plates were incubated at 37 °C and 5% CO_2_. 

Approximately 48 h after transduction, the ONE-Step Luciferase reagent was prepared per the recommended protocol. One hundred microliters of ONE-Step Luciferase reagent were added per well and the plate was rocked at room temperature for ~30 min. The luminescence was measured using a luminometer (BioTek Synergy^TM^ 2 microplate reader; Agilent, Santa Clara, CA, USA). 

### 2.4. Data Analysis

The reporter assays were performed in triplicate at each concentration. The luminescence intensity data were analyzed using the computer software GraphPad Prism (GraphPad Software, Boston, MA, USA). In the absence of the compound, the luminescence intensity (L_t_) in each data set was defined as 100%. In the absence of pseudovirus, the luminescence intensity (L_b_) in each data set was defined as 0%. The percent of luminescence in the presence of each compound was calculated according to the following equation: % Luminescence = (L − L_b_)/(L_t_ − L_b_), where L = the luminescence intensity in the presence of the compound, L_b_ = the luminescence intensity in the absence of virus, and L_t_ = the luminescence intensity in the absence of the compound. 

## 3. Results and Discussion

The chemical composition of Peptide 1 is SLDQINVTFLDLEYEMKKLEEAIKKLEES- YIDLKELK(Peg_5_)-NH_2_ and that of Peptide 2 is SLDQINVTFLDLEYEMKKLEEAIKKLEES-YIDLKELK(Palm-Peg_5_)-NH_2_. 

Peptide 1 has the same amino acid sequence as the EK1 peptide of Xia et al. [5], except for the lysine linker at the C-terminus as well as a 17-amino-3,6,9,12,15-pentaoxaheptadecanoyl moiety (PEG_5_) moiety to render it a more appropriate control for the palmitoylated Peptide 2. 

In our initial experiment, we examined the effects of Peptide 1 and Peptide 2 on SARS-CoV-2 S-protein-mediated fusion. Peptide 1 and Peptide 2 had no inhibitory effect at concentrations of 0.1 to 1 µM (Figure 4). This result was surprising in view of the nanomolar inhibitory concentration of EK1 reported by Xia et al. [5] in an analogous system.

We then examined higher concentrations of the peptides. Peptide 1 had 25% inhibitory activity at 10 µM, whereas Peptide 2 inhibited fusion by 76% at the same concentration (Figure 5).

These inhibitory effects are much lower than those reported by Xia et al. [5] for the inhibition of cell–cell fusion. In an assay measuring SARS-CoV-2-mediated cell–cell fusion utilizing 293T/S/GFP effector cells and Huh-7 target cells, these authors found an IC_50_ of 287 nM for EK1 (corresponding to our Peptide 1) and an IC_50_ of 69 nM for EK1P (corresponding to Peptide 2). Their palmitoylated EK1P construct had a spacer of PEG4 instead of our PEG_5_. It is possible that small variations in the derivatization of the HR2-based peptides may affect the antiviral activity of the peptides.

In a pseudovirus fusion assay, the IC_50_ of EK1 was found to be 2.375 µM [5]; i.e., much higher than that found in the cell–cell fusion assay. Thus, it may be reasonable to expect that in our assay for SARS-CoV-2 S-pseudovirus fusion, the inhibition of fusion by the peptides may require higher concentrations than in a cell–cell fusion assay. Zhu et al. [7] also observed much higher IC_50_s for their HR1-derived peptides in the pseudovirus assay than in their cell–cell fusion assay. It is also of interest to note that Xia et al. [5] observed only a 60% inhibition of pseudovirus fusion by EK1 at 10 µM. Our Peptide 1, however, inhibited infectivity by only 25% at 10 µM.

It is relevant here to point out previous observations of the differences between the inhibitory effects of peptides or antibodies on HIV-1-mediated cell–cell fusion vs. virus infectivity [8,9].

In addition to the potential effects of small variations in peptide derivatization, it is likely that there are differences in the S-protein density in the pseudovirus membranes, resulting from variations in the pseudovirus preparation methodologies between laboratories.

We suggest that future studies on the inhibition of infectivity of SARS-CoV-2 in both in vitro and in vivo systems consider the need for higher concentrations of peptide inhibitors.

## Figures and Tables

**Figure 1 ijms-25-06382-f001:**
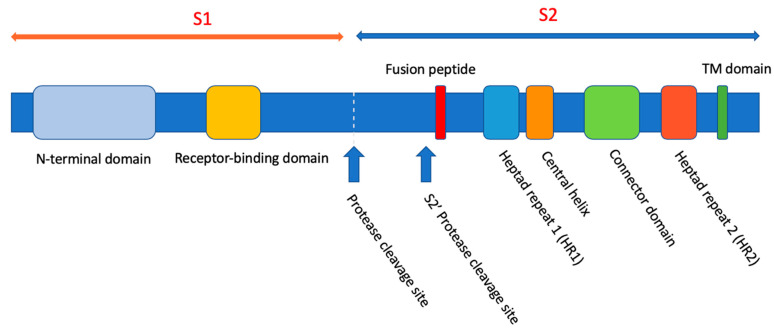
The domains of the SARS-CoV and SARS-CoV-2 spike protein, S. Reproduced from [2].

**Figure 2 ijms-25-06382-f002:**
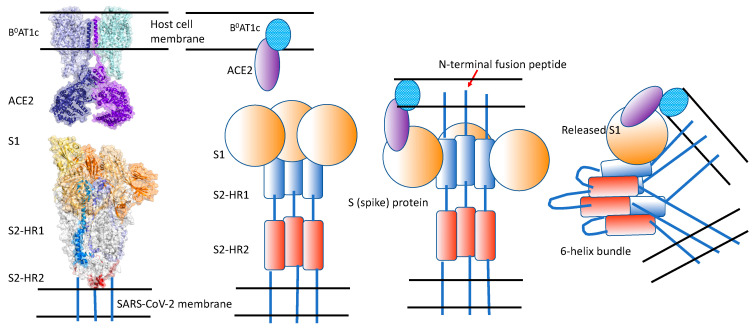
The conformational change in the SARS-CoV-2 spike protein and formation of the “six-helix bundle”. Reproduced from [1].

**Figure 3 ijms-25-06382-f003:**
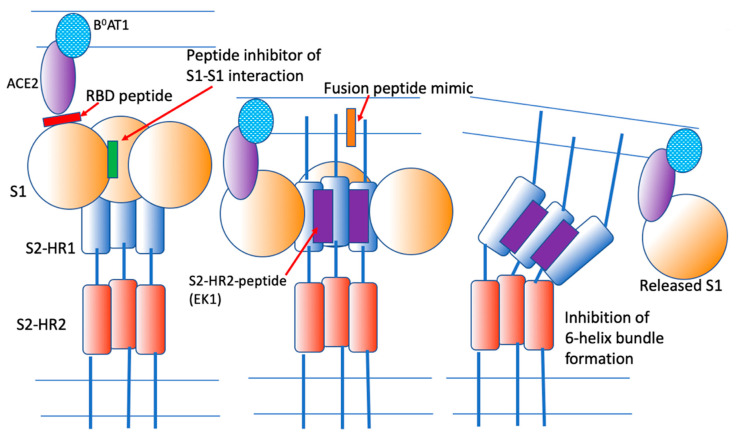
Inhibition of membrane fusion by various peptides. Red: A peptide that binds to the receptor-binding domain of S1. Green: A peptide that inhibits the interaction between S1 subunits. Orange: An S2 fusion peptide mimic that may inhibit the interaction of the fusion peptide with the target membrane. Purple: A peptide derived from the S2-HR2 region that binds with high affinity to the HR1 region and inhibits the interaction between the S2-HR1 and S2-HR2 domains, thus preventing “six-helix bundle” formation and membrane fusion. Reproduced from [1].

**Figure 4 ijms-25-06382-f004:**
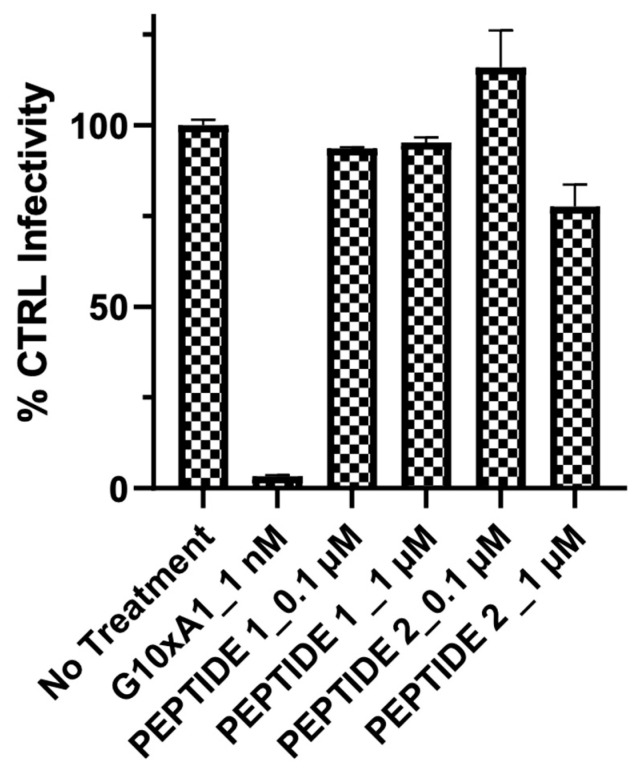
The effect of 0.1 µM and 1 µM Peptide 1 and Peptide 2 on the infectivity of the pseudotyped vesicular stomatitis virus expressing the SARS-CoV-2 S-protein. G10xA1 is a neutralizing antibody against the spike protein S1.

**Figure 5 ijms-25-06382-f005:**
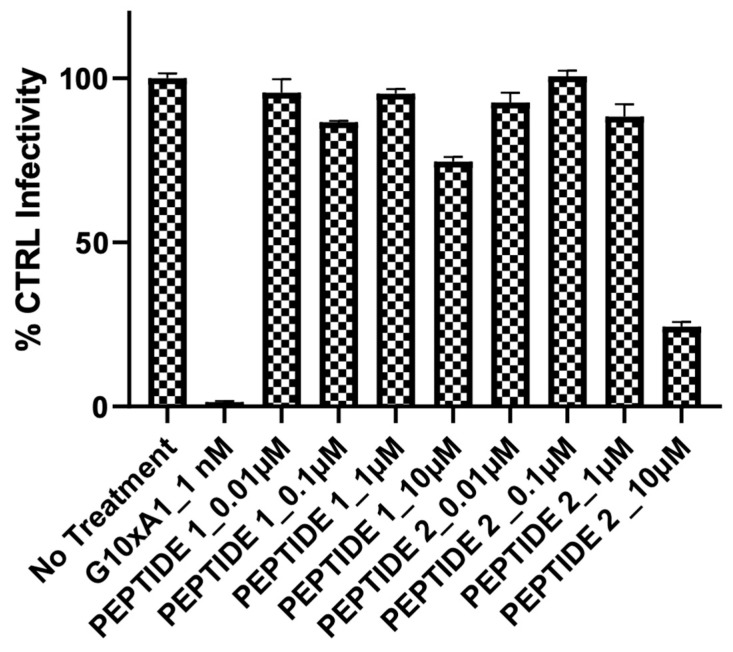
The effect of Peptide 1 and Peptide 2 in the range of 0.01 µM to 10 µM on the infectivity of the pseudotyped vesicular stomatitis virus expressing the SARS-CoV-2 S-protein.

## Data Availability

All of the obtained data are presented in this publication.

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
