# Peer review of "Peptides Derived from the SARS-CoV-2 S2-Protein Heptad-Repeat-2 Inhibit Pseudoviral Fusion at Micromolar Concentrations: The Role of Palmitic Acid Conjugation"

_ijms, 2024, doi:10.3390/ijms25126382_

Round 1

Reviewer 1 Report

Comments and Suggestions for Authors

The authors tested two derivatives of SARS-CoV-2 HR2 peptide for their ability to inhibit fusion mediated by vesicular stomatitis virus pseudotyped with SARS-Cov-2 spike. The authors tested an HR2 peptide that was derivatized by palmitic acid conjugation. While the results are clear and the data well presented, the authors fail to convey why this research was done and the significance of their findings. Thus, the enthusiasm for this work is low. In addition, some key controls are missing.

1.  Line 27: there is a formatting error

2. The authors do not justify why they derivatized a previously used HR2 peptide with palmitic acid. What is the biological or mechanistic reason for testing this derivative? Without this explanation the significance of this paper is lacking.

3. The authors missed an important control by not using the exact HR2 peptide that was used in the previous publication. Without that control it is not clear if the confounding results they observe are due to small variations in the HR2 peptide, or due to differences in their assay (cell fusion vs. pseudotyped vaccinia virus).

4.  The Discussion needs to be improved by restating the purpose of these set of experiments and linking palmitic acid conjugation to the biology and mechanism of fusion.

5. Although the title mentions "Role of palmitic acid conjugation", the introduction, results and discussion do not address the role of palmitic acid conjugation in the fusion mechanism in any meaningful way.

Author Response

We appreciate the insightful comments of the Reviewer.

We have modified the Introduction to explain why this research was done. We were intrigued by the very low IC50 values obtained by Xia et al. in their cell-cell fusion system, and wanted to examine the potential inhibitory activities of the two peptides in a virus-cell fusion system.

  1. We have corrected line 27.
  2. We explain the rationale for the use of palmitic acid in the revised manuscript. First, Xia et al. used this derivatization; thus, we wanted to compare their results with ours. Second, we hypothesized that the palmitic acid would anchor the peptide in the viral and cell membranes, and thus increase the local concentration of the peptide, thereby facilitating its interaction with the HR1 region of the S protein.
  3. Because of budget limitations, we chose to synthesize Peptide 1 with the PEG5 spacer to act as an appropriate control for Peptide 2 with the conjugated palmitic acid that required the PEG5 spacer. Regarding the potential differences in inhibition of cell-cell fusion vs pseudovirus infectivity, we have already pointed out the results of Zhu et al (2020) and Xia et al. (2020b). Thus, it is not unexpected that pseudovirus infectivity requires higher inhibitory concentrations of peptide. In this respect, we have also cited our previous studies on differences in inhibitory concentrations of peptides and antibodies in cell-cell fusion vs vfiral infectivity systems (Konopka et al., 1995a, 1995b).
  4. and 5. We have added a discussion of the potential mechanism of action of palmitic acid in enhancing the inhibitory effect of Peptide 1.

Reviewer 2 Report

Comments and Suggestions for Authors

This study aimed to verify the efficacy of a previously reported peptide that targets the class one fusion machinery mechanism of SARS-CoV-2 spike protein, within the S2 heptad repeat. The authors also slightly modified the previously reported peptide to determine any enhancement of neutralisation effect. 

Overall, i am very 50/50 with this brief report. Its true that there are differences between pseudotype virus preparations, though until this recent study by Cantoni et al, there appears to be a high correlation between live-virus and pseudovirus, which would then also indicate agreement between pseudotype assays from different labs. PMID: 37790941. I will leave it to author's discretion whether this is worth citing, but it may put into question the statement made at the end regarding spike density.

One key difference that authors should be mentioning is possibly the titre of pseudotype virus that can differ greatly between laboratories due to many reasons: cells, passage number, etc. Many labs titre their PV using the reporter, which is not a standardised method, since plate readers, even the same models, can return different values for the same preparation of virus. In any case, its clear that the peptide in question does not appear anywhere near as potent as the other paper had suggested.

I believe that the authors of this study should have really carried out a neutralisation experiment, with curves, to accurately determine the IC50, as described in this methods protocol: PMID: 31164554. are the authors able to do this?

i look forward to the modifications to the manuscript.

Comments on the Quality of English Language

some minor typo:

line 64: ptotein should be protein.

Author Response

We apprecite the insightful comments of the Reviewer.

  1. Regarding the potential differences in inhibition of cell-cell fusion vs pseudovirus infectivity, we have already pointed out the results of Zhu et al (2020) and Xia et al. (2020b). Thus, it is not unexpected that pseudovirus infectivity requires higher inhibitory concentrations of peptide. In this respect, we have also cited our previous studies on differences in inhibitory concentrations of peptides and antibodies in cell-cell fusion vs vfiral infectivity systems (Konopka et al., 1995a, 1995b).
  2. Another factor contributing to differences in the inhibitory effect of peptides in different laboratories is the variations in pseudovirus preparation methodologies. We have indicated this in the revised manuscript.
  3. We have added a comment at the end of the manuscript and the abstract regarding the potential need for higher concentrations of peptide inhibitors in future studies on in vitro and in vivo infectivity of SARS-CoV-2.

Round 2

Reviewer 1 Report

Comments and Suggestions for Authors

The authors have addressed my concerns, and I am satisfied with their responses.

Reviewer 2 Report

Comments and Suggestions for Authors

Authors addressed my comments.